# Experience in the Cultivation of a New Perennial Cereal Crop—Trititrigia in the Conditions of South of the Rostov Region

Yuri Lachuga [1], Besarion Meskhi [2], Viktor Pakhomov [3,4], Yulia Semenikhina [3], Sergey Kambulov [3,4], Dmitry Rudoy [4,*] and Tatyana Maltseva [4]

1. Russian Academy of Sciences (RAS), Russian Federation, Leninsky Avenue, 32a, 119334 Moscow, Russia
2. Department of Rector, Don State Technical University, 1, Gagarin Sq., 344003 Rostov-on-Don, Russia
3. FSBSI Agricultural Research Center "Donskoy", Nauchny Gorodok Str., 3, 347740 Zernograd, Russia
4. Faculty of Agribusiness, Don State Technical University, 1, Gagarin Sq., 344003 Rostov-on-Don, Russia
* Correspondence: dmitriyrudoi@gmail.com

**Abstract:** The article presents the research results on the cultivation of a new perennial cereal crop of winter wheat hybrid and wheatgrass—perennial winter wheat (Trititrigia) of the "Pamyati Lyubimovoy" variety (hereinafter—Trititrigia) in the southern zone of the Rostov region over two years. The purpose of the research is to assess the degree of suitability for the use of a new perennial cereal crop—Trititrigia in the southern zone of the Rostov region. The yield, technological indicators of grain, and baking properties of flour were taken as evaluation criteria. The study of all aspects of Trititrigia cultivation was carried out in comparative sowings with winter wheat of the Stanichnaya variety, common in the Rostov region, of the Agricultural Research Center "Donskoy" (ARC Donskoy). As a result of the research, it was found that the average biological yield of Trititrigia in the southern zone of the Rostov region in two years was 4.28 t ha$^{-1}$, which was 1.57 t ha$^{-1}$ less than that of the control sowing of winter wheat of the Stanichnaya variety. The weight of the straw part with an ear of Trititrigia is 1.9 times higher than that of winter wheat of the "Stanichnaya" variety. Technological indicators of the quality of Trititrigia grain corresponded to the first class in terms of amount of protein (more than 19%), gluten (33.34%), and falling number (274 s); the third class according to the gluten deformation index (GDI) (81.5 points); the fifth class according to the natural mass of grain (691 g L$^{-1}$). The general baking evaluation of Trititrigia grain allowed it to be classified as valuable wheat.

**Keywords:** Trititrigia; perennial crop; wheat; cultivation; grain quality; baking evaluation





## 1. Introduction

It is known that the idea of creating perennial wheat belongs to Nikolai Tsitsin and originates from the 1920s–1930s of the last century [1]. He also owns the first attempt to create it by crossing wheat with wheatgrass [2]. Subsequently, this idea was supported and developed by scientists in Canada, the USA, Germany, and China [3,4]. Now there are already extensive studies of wheat–couch grass hybrids of domestic and foreign breeders [5–7]. Now scientists have access to modern methods of genomic engineering and molecular selection [8–11], which allow the development and study of various breeding materials within a short period of research [12].

The main strategies for obtaining wheat–couch grass hybrids are interspecific and intergeneric hybridization [5,13] between cultivated and wild plants [12,14], in particular, between common annual wheat and its perennial wild relative, wheatgrass [15–17]. As a result, the new plant combines the perennial features of wheatgrass growth with the quality of wheat grain [18] and also has ecological plasticity, resistance to adverse environmental factors, diseases, and other valuable traits and properties [6,19].

The advantage of perennial cereal crops lies in the multifunctionality of their application [20,21]. Such plants provide a continuous soil cover and have a longer growing

season [22] than annual crops. They also have a developed root system, which increases their resistance to drought [23,24] and allows efficient use of moisture in soil. In ecological terms, perennial crops can act as a soil protection mechanism [25,26], prevent soil erosion and deflation, and preserve the soil microbiome [27,28]. Economic benefits of using perennial crops include reduced costs for seeds, tillage, and other activities, i.e., production costs are reduced overall.

However, despite the listed advantages of perennial wheat, now it is not of serious interest to a modern producer of grain crops [29,30]. Firstly because of the lack of extensive and convincing studies on its cultivation in various soil and climatic conditions. Implementation of a perennial crop will require a revision of crop rotations and sown areas [31,32]. Not every producer is ready to go for the reconstruction of his crop production [30]. Most importantly, today it is not clear if the crop of perennial wheat is in demand.

It is important to conduct integrated studies of crop breeders, agronomists, and engineers to determine the viability, scope, adaptability of perennial wheat breeding varieties in various soil and climatic conditions [33,34].

Russian crop breeders, developing the idea of Tsitsin V.N. for a long period of time, carried out breeding work on distant hybridization of wheat with two species of wild wheatgrass Elytrigia intermedium (Thinopyrum intermedium (Host) Barkworth and D.R. Dewey) and E. elongate (T. elongatum (Host) Dewey). As a result, they obtained a fundamentally new crop, perennial wheat, which later became known as Trititrigia (Trititrigia cziczinii Tsvelev) [35].

In 2020, Trititrigia was included as a separate agricultural crop in the State Register of Breeding Achievements Approved for Use in the Russian Federation [36,37]. The originator and patent holder of this variety is the Federal State Budgetary Institution of Science, the Main Botanical Garden named after N.V. Tsitsin of the Russian Academy of Sciences.

Through the hybridization of wheat with wheatgrass, authors of the new culture solved breeding problems to create valuable perennial and growing forms of cereals [38]. As a result, the Trititrigia genome was obtained, represented by 56 chromosomes (42 from wheat and 14 from wheatgrass). Breeding crop plants combine the following unique traits from parental species [39,40]: winter type of development, productive high bushiness, and strong stem; intensive growth of shoots after ripening and harvesting of grain, provoking up to three cuttings of the green mass crop; high rates of winter hardiness and drought resistance; highly resistance to leaf rust, powdery mildew, septoria, and head blight with moderate susceptibility to yellow rust, long growing season; relative resistance to lodging at a plant height of 135–150 cm; the ear is cylindrical, 10–15 cm long, loose, white, awnless; caryopsis of medium size, colored, elongated oval. It is immune to dusty and hard smut. The mass of 1000 grains is 31–35 g. The grain of Trititrigia has high-quality indicators: protein 18–19%, crude gluten 42.7–43.1% with grain size 774–800 g L$^{-1}$. High baking score from 3.8 to 4.0 points [41].

Another long-term grain crop was obtained by selection: intermediate wheatgrass (IWG) under the trademark Kernza (The Land Institute, Saline, KS, USA), derived from wheatgrass [42]. The study of the effect of intermediate wheatgrass (IWG) perennial grain (Kernza) crops on reducing the number of weeds was carried out by scientists [42]. The crops were sown in the southeast of France. The article also presents the results of grain yields, which amounted to 8.99 and 8.54 t ha$^{-1}$. Scientists note that grain yields were lower during the first year and decreased significantly after the second harvest. In [43], the properties of the dough obtained from intermediate wheatgrass (IWG) flour were investigated. The results showed differences in the properties of the protein of ordinary wheat and IWG: proteins in IWG have high solubility, on the basis of which scientists assume that the protein network is based more on non-covalent interactions.

Scientists of the N.V. Tsitsin Main Botanical Garden were engaged in research on the cultivation of Trititrigia [36,37,44]. Trititrigia was grown in the Moscow region, whose climate is significantly different from the Rostov region: the Rostov region is characterized by a more arid climate and high temperatures in summer and winter. In studies [36,37,44]

Trititrigia was grown on experimental plots with sod-podzolic heavy loamy soil, while in the Rostov region, ordinary carbonate heavy loamy chernozem prevails.

The above-mentioned economically valuable signs demonstrate the extensive potential of Trititrigia, which served as the main motivation for studying its cultivation in the arid conditions of Southern Russia with insufficient and unstable moisture.

From an agronomic point of view, it is advisable to obtain an objective assessment of the adaptability and stability of Trititrigia when conducting test tests similar in nature to environmental ones. Ecological tests of bred crops are carried out in various soil and climatic zones to assess the response of crops to changes in growing conditions or environmental sustainability and to determine the recommended cultivation regions for the studied varieties.

In this connection, the test (ecological) test of Trititrigia is very relevant, since it allows us to identify the effectiveness of perennial wheat by the main economically valuable characteristics, objectively assess the ecological stability and stability of a new crop by its productivity in the given soil and climatic conditions of the South of Russia.

Therefore, the purpose of this study is to test a new crop (Trititrigia) in the semi-arid southern zone of the Rostov region, grown in the traditional cultivation system by identifying its adaptability according to yield criteria, technological indicators of grain and baking properties of flour in comparative crops with local winter wheat of the "Stanichnaya" variety.

## 2. Materials and Methods

### 2.1. Territory Experimental Field

The southern zone of the Rostov region belongs to the zone of risky agriculture with insufficient and unstable moisture. It is characterized by a semi-arid climate with hot summers and moderately cold winters. On average, 488.5 mm of precipitation falls per year, the average annual temperature is about 9.7 °C, the hydrothermal coefficient is 0.7–0.8. The duration of the frost-free period is 180–210 days. The average number of dry days per year reaches 80–85 [45].

The research of Trititrigia cultivation was established on the experimental field of the Agricultural Research Center "Donskoy" (Russian Federation, Rostov Region, city of Zernograd, N 46.8124°, E 40.3036°). Experimental area (270 m$^2$) for Trititrigia crops was placed on the territory of long-term stationary (Figure 1).

**Figure 1.** The location scheme of the experimental sowing of Trititrigia in the conditions of long-term stationary experience of cultivation according to the traditional system.

The soil of the experimental site is ordinary carbonate heavy loamy chernozem (Voronic Chernozems Pachic according to WRB-2014). The main indicators of fertility in the arable layer (0–30 cm) of the soil are shown in Table 1.

**Table 1.** Indicators of the soil fertility in the experimental area.

| Bulk Density (g cm$^{-3}$) | pH | Humus (%) | Content Total N (mg kg$^{-1}$) | Available Phosphorus (mg kg$^{-1}$) | Available Potassium (mg kg$^{-1}$) |
|---|---|---|---|---|---|
| 1.25 | 7.1 | 3.3 | 28.2 | 19.0–24.5 | 327–337 |

The bulk density of the soil was measured using the drilling method by taking a soil sample of undisturbed addition using a cylinder drill. The pH of the soil was measured by potentiometric method, i.e., by displacing hydrogen cations from the soil-absorbing complex using a hydrolytically neutral potassium chloride salt and determining the concentration of hydrogen cations in the resulting solution using a pH meter. Humus was determined according to the method of I.V. Tyurin with modification of V.N. Simakov, using a photoelectrocolorimeter after oxidation of organic matter with a solution of potassium bicarbonate in sulfuric acid, followed by determination of trivalent chromium. Content total nitrogen (N) was determined using the method of Kjeldahl, consisting of decomposition of soil organic matter with concentrated sulfuric acid during boiling and subsequent quantitative accounting of the resulting nitrogen. Available phosphorus and potassium in the soil were determined through method of Kirsanov with modification of (Central Institute of Agricultural Chemistry Service, Moscow, RF) by extraction of mobile phosphorus compounds ($P_2O_5$) and potassium ($K_2O$) from the soil with hydrochloric acid solution and subsequent quantitative determination of mobile phosphorus compounds on a photoelectrocolorimeter and potassium on a flame photometer.

*2.2. Soil Preparation*

In the traditional system of cultivation of agricultural crops, mechanized tillage is carried out. Soil preparation before sowing Trititrigia was carried out with a combined KUM-4 unit developed by the ARC "Donskoy" (Figure 2). This unit is designed for shallow layer-by-layer tillage to a depth of 16 cm with simultaneous cutting of weeds, crumbling blocks, leveling, compacting, and mulching the surface soil layer.

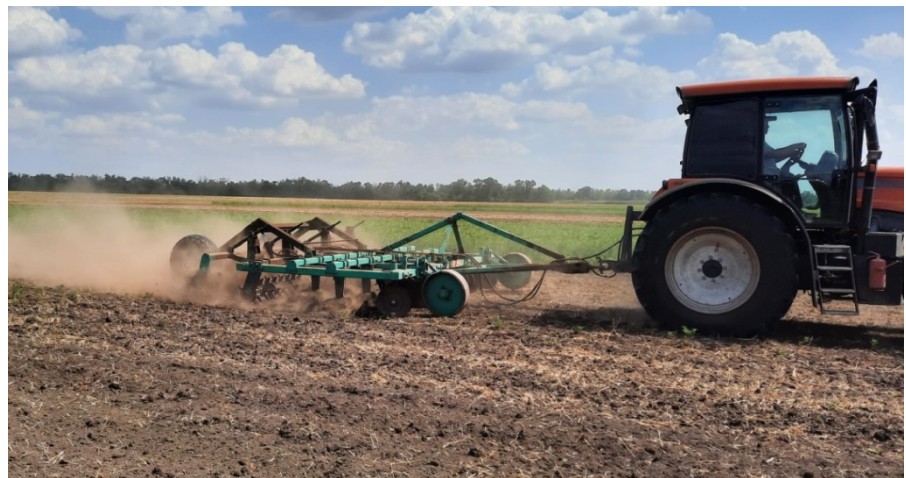

**Figure 2.** Soil preparation by combined unit KUM-4.

Due to the rational combination of tillage operations of the KUM-4 unit, it is possible to prepare the soil to the pre-sowing state in one pass across the field. Leveled (smooth) bottom of the furrow, separated soil, and alternation of loose and compacted soil layers

provide not only the preservation of residual soil moisture but also its accumulation through condensation during the dry period [46,47].

*2.3. Sowing Fertilizers*

Seed material for testing perennial wheat in the southern zone of the Rostov region was provided by the seed originator—the N. V. Tsitsin Main Botanical Garden of the Russian Academy of Sciences (Moscow, Russian Federation). Seeds of winter wheat "Stanichnaya", used in the experiment as a control, were bred by the Agrarian Scientific Center "Donskoy". Both varieties have drought resistance and frost resistance.

Trititrigia seeds were sown with the "Demetra" selection seeder developed by the ASC "Donskoy" (Figure 3). This seeder performs precise sowing of seeds thanks to cone sowing units that distribute a portion of seeds evenly along the length of the row [48]. The sowing depth of seeds is 6–7 cm, the row spacing is 15 cm, the seeding rate of Trititrigia seeds is 150 kg ha$^{-1}$.

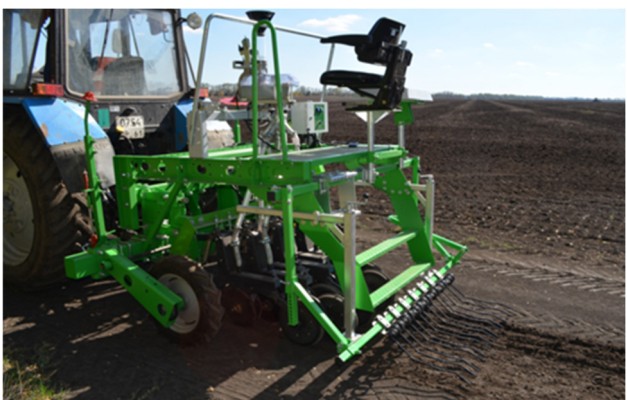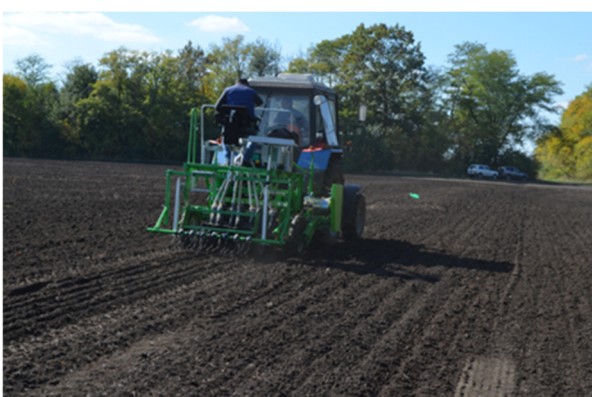

**Figure 3.** Trititrigia sowing with the Demetra seeder.

Trititrigia seeds were sown with a recommended seeding rate of 4.3 million pcs ha (150 kg ha$^{-1}$). Seeds of winter wheat "Stanichnaya" were sown with the recommended seeding rate of 5.5 million pcs ha$^{-1}$ (220 kg ha$^{-1}$).

In 2020, seeds of Trititrigia and winter wheat were sown on 5 October. The first shoots of Trititrigia appeared simultaneously with winter wheat—on 18–19 October 2020, in March 2021, both crops resumed their vegetation. The growing season of Trititrigia lasted until 4 August 2021 and amounted to 304 days. The growing season of winter wheat lasted 288 days until 19 July 2021.

In 2021, Trititrigia and winter wheat were sown on 24 September. The first shoots of the tested plants appeared on 11–12 November. The delay in germination was due to the small amount of precipitation in September–October. The resumption of vegetation of both crops occurred in March 2022. The growing season of Trititrigia lasted until 2 August 2022 and amounted to 276 days. The growing season of winter wheat, lasting 261 days, lasted until 11 July 2022.

In total, three fertilizing feedings were carried out during the vegetation period of the plants. The first application of the ammophos fertilizer at a dose of 100 kg ha$^{-1}$ was carried out simultaneously with the sowing of seeds of Trititrigia and winter wheat. Subsequent second and third fertilizing was carried out with ammonium nitrate at a dose of 70 kg ha$^{-1}$ in the phase of spring tillering and in the phase of plants entering the tube for each test culture.

Observations of plants and their records were carried out in accordance with the Methodology of State Variety Testing of Agricultural Crops [49].

### 2.4. Weather and Climatic Conditions

For the period of the Trititrigia cultivation experiment, meteorological parameters were taken into account, which included temperature and precipitation (Figure 4). According to the prevailing meteorological conditions, the vegetation period of plants proceeded mainly in arid conditions.

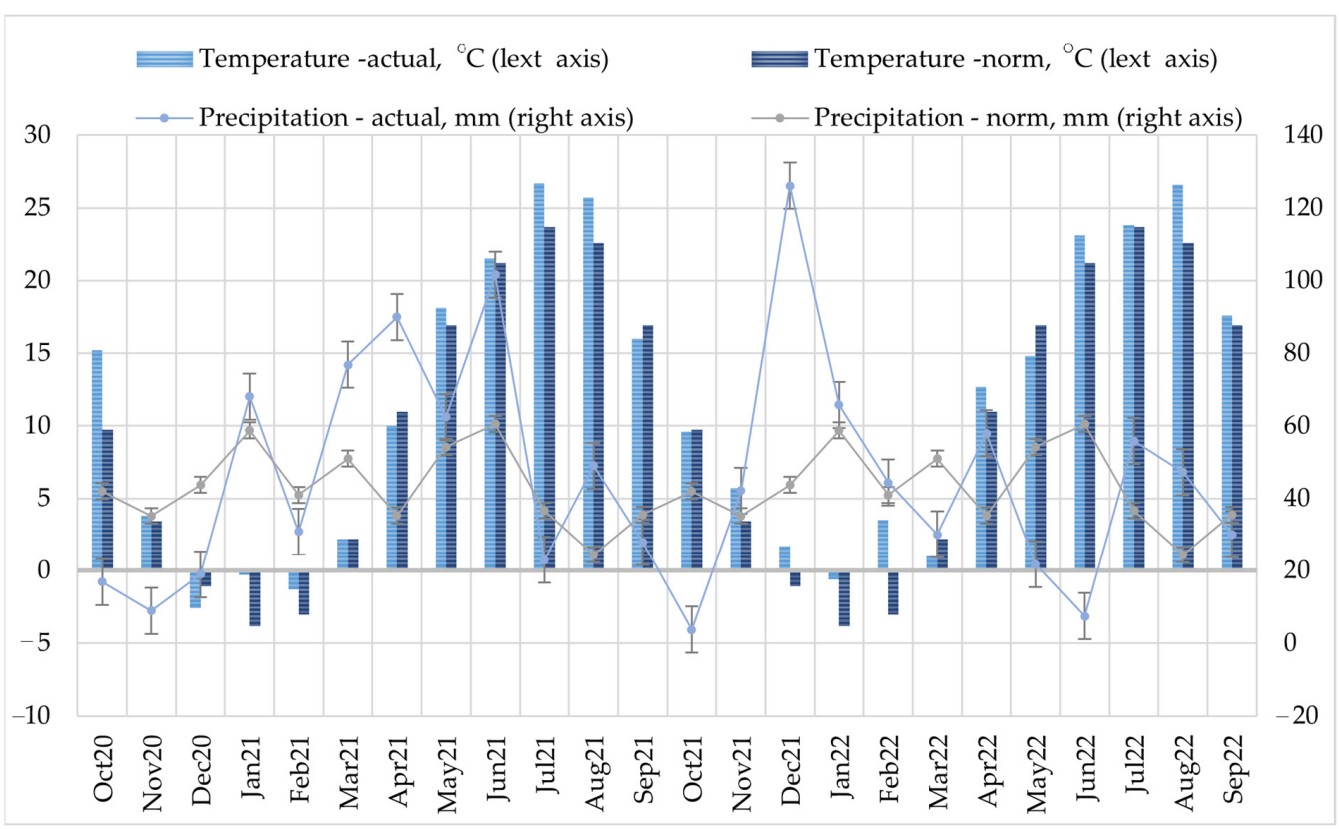

**Figure 4.** Meteorological parameters: temperature and precipitation during the test period.

For the observed period corresponding to the growing season of Trititrigia, from October 2020 to August 2021, the meteorological conditions were as follows. In the first three months—October, November, and December—there was a significant shortage of them compared to the average multi-year norm. During this period, 44.8 mm of precipitation fell at a rate of 120.9 mm. However, from January until the end of the growing season, the amount of precipitation exceeded the average long-term norm and amounted to 541.3 mm by the end of the growing season. The highest excess of precipitation norms was observed in March, April, June, and August. The maximum excess of monthly norms was 150% in March, 255% in April, 115% in May, 168% in June, and 179% in August. The temperature conditions during this period were as follows. After sowing, a decrease in air temperature was observed. The average monthly temperature from October to December decreased from +15.2 °C in October to +3.8 °C in November and to −2.6 °C in December. In winter, the temperature remained at 0 °C in January and −1.3 °C in February. At the same time, the minimum of minimum temperature was recorded in January at −21.4 °C, and the minimum of maximum temperature was +6.8 °C in December. From March to July, there was an increase in air temperatures. The average monthly temperature increased from +2.2 °C in March to +26.7 °C in July and the maximum values from +14.3 °C in March to +39.5 °C in July. The minimum temperatures in the spring–summer growing season of Trititrigia were −12.9 °C in March, +2.2 °C in April, +5.1 °C in May, +11.3 °C in June, and +16.7 °C in July. In the final month of August, temperatures slightly decreased: the monthly average was +25.7 °C, the maximum was +37.3 °C, and the minimum was −15.9 °C.



During the growing season of Trititrigia from September 2021 to August 2022, meteorological conditions were contrasting. In the autumn of 2021, there was a shortage of precipitation by 57.8 mm (from the norm of 131.5 mm). In winter, precipitation fell by 90.5 mm more than normal (145.7 mm), while a significant excess in the number of precipitations was recorded in December by 62.8 mm (norm 63.3 mm), in January by 20.7 mm (norm 45.1 mm) and in February by 7.0 mm (norm 37.3 mm). In the spring, precipitation fell by 21.4 mm less than the average annual data (the norm is 131 mm), and in the summer precipitation fell by 63.8 mm less than the norm (174.2 mm). As a result of the analysis of the recorded data on the average daily air temperature, it was revealed that in the autumn of 2021, high air temperatures were observed, which were higher than the average annual by 0.7 °C (from the norm of 9.7 °C). Winter was warm relative to the average long-term temperatures by 1.2 °C (with a norm of −2.67 °C), while December was warm, when there was an increase in temperature by 2.9 °C relative to the average long-term norm (−1.2 °C), in January it was warmer by 3.2 °C (norm −3.8 °C). It was especially warm in February, when there was an increase in temperature by 6.5 °C relative to the average annual norm (−3.0 °C). In spring, the air temperature remained at the level of the average annual and exceeded the average annual norm (9.5 °C) by 0.2 °C. The summer was hot, with an excess of the average annual norm (21.8 °C) by 2.7 °C, while June and August were especially hot, when there was an increase in air temperature by 2.6 °C and 4.7 °C, respectively (with a norm of 20.5 °C and 21.9 °C).

*2.5. Harvest*

Before harvesting Trititrigia, its biological yield was determined through studies of the elements of the crop structure. Yield of grain crops depends on the productivity of each plant, their number per unit area, the number of grains in an ear, and the mass of 1000 grains.

To determine the biological yield, sheaves of plants were taken from the accounting plot in quadruple replication and studied under laboratory conditions. Despite the fact that the biological yield often does not coincide with the combined (actual), nevertheless, its definition gives a preliminary idea of the magnitude of the yield and, most importantly, of its constituent elements and also allows evaluation of the valuable economic characteristics of the cultivated crop.

*2.6. Baking Properties of Flour*

The baking properties of Trititrigia grain flour and winter wheat were determined during trial laboratory baking of bread by the remix method with repeated kneading. The rheological properties of the test were determined according to GOST R 51415-99 (ISO 5530-4-91) using an alveograph.

Descriptive statistics with the determination of the average linear deviation and coefficient of variation were used to process the results of the study.

**3. Results**

*3.1. Elements of Yield Structure*

The obtained results of the indicators of the elements of the Trititrigia crop structure in comparison with the annual winter wheat of the Stanichnaya variety are presented in Table 2.

As a result of the analysis of the data in Table 1, it was found that Trititrigia plants are tall at 140.39 cm, and have a long ear of 15.37 cm, as a result of which a high yield of straw mass of 1595.04 g m$^{-2}$ develops, which by 67.40% prevails over the mass of winter wheat straw, since the height of wheat plants is 92.75 cm, and the length of the ear is 8.11 cm.



**Table 2.** Trititrigia and winter wheat crop structure elements.

| Indicators | Years | Trititrigia "Pamyati Luybimovoy" | | | Winter Wheat "Stanichnaya" | | |
|---|---|---|---|---|---|---|---|
| | | Arithmetical Mean Value (Mean) | Average Linear Deviation (ALD) | Coefficient of Variation (CV) | Arithmetical Mean Value (Mean) | Average Linear Deviation (ALD) | Coefficient of Variation (CV) |
| Plant height, cm | 2021 | 141.43 | 7.57 | 7.18 | 94.17 | 1.86 | 2.55 |
| | 2022 | 139.34 | 1.61 | 1.70 | 91.33 | 2.06 | 3.00 |
| | Average | 140.39 | 4.10 | 3.41 | 92.75 | 1.54 | 1.98 |
| Number of plants with root, pcs. | 2021 | 119.08 | 2.02 | 3.50 | 191.33 | 90.00 | 5.78 |
| | 2022 | 113.17 | 2.33 | 2.52 | 186.17 | 17.83 | 14.53 |
| | Average | 116.13 | 2.04 | 2.24 | 188.75 | 9.63 | 6.81 |
| Ear length, cm | 2021 | 15.55 | 0.64 | 6.45 | 7.81 | 0.54 | 8.24 |
| | 2022 | 15.20 | 0.36 | 3.83 | 8.42 | 0.40 | 5.61 |
| | Average | 15.37 | 0.47 | 3.66 | 8.11 | 0.35 | 5.16 |
| The mass of the straw part with an ear, g | 2021 | 2030.30 | 55.59 | 3.47 | 976.40 | 10.30 | 1.82 |
| | 2022 | 1962.18 | 11.75 | 2.90 | 1060.96 | 131.77 | 23.63 |
| | Average | 1996.24 | 40.63 | 2.52 | 1018.68 | 66.71 | 12.41 |
| Number of productive stems, pcs. | 2021 | 416.08 | 7.36 | 3.26 | 396.35 | 7.18 | 2.10 |
| | 2022 | 387.33 | 9.52 | 4.33 | 408.32 | 13.14 | 4.02 |
| | Average | 401.71 | 5.79 | 2.12 | 402.34 | 7.37 | 2.25 |
| Weight of pure grain from ears, g | 2021 | 409.47 | 3.74 | 1.31 | 485.00 | 14.14 | 3.83 |
| | 2022 | 392.92 | 5.76 | 3.05 | 512.42 | 15.22 | 3.80 |
| | Average | 401.20 | 4.58 | 1.41 | 498.71 | 8.80 | 2.63 |
| Weight of one ear, g | 2021 | 0.99 | 3.69 | 0.02 | 1.23 | 0.07 | 8.98 |
| | 2022 | 1.02 | 5.21 | 0.04 | 1.64 | 0.12 | 9.89 |
| | Average | 1.00 | 2.88 | 0.02 | 1.44 | 0.08 | 7.96 |
| Mass of 1000 grains, g | 2021 | 24.76 | 0.54 | 3.45 | 41.61 | 1.62 | 5.65 |
| | 2022 | 22.40 | 0.63 | 3.09 | 44.36 | 2.61 | 7.17 |
| | Average | 23.58 | 0.42 | 2.32 | 42.98 | 1.79 | 5.65 |
| Number of grains per ear, pcs. | 2021 | 39.75 | 1.21 | 3.74 | 29.49 | 1.94 | 8.70 |
| | 2022 | 45.31 | 1.20 | 3.25 | 28.50 | 2.35 | 10.61 |
| | Average | 42.55 | 0.75 | 2.29 | 29.02 | 1.92 | 8.85 |
| Straw mass, g m$^{-2}$ | 2021 | 1620.53 | 54.43 | 4.38 | 491.40 | 11.91 | 3.56 |
| | 2022 | 1569.25 | 14.02 | 3.45 | 548.54 | 133.68 | 46.32 |
| | Average | 1595.04 | 29.90 | 3.13 | 519.97 | 66.83 | 24.25 |
| Ratio straw/grain | 2021 | 3.96 | 0.17 | 4.75 | 1.01 | 0.04 | 6.40 |
| | 2022 | 4.00 | 0.05 | 4.14 | 1.08 | 0.27 | 47.90 |
| | Average | 3.98 | 0.10 | 3.32 | 1.05 | 0.13 | 24.56 |
| Biological yield, t ha$^{-1}$ | 2021 | 4.38 | 0.05 | 1.68 | 4.99 | 0.29 | 9.58 |
| | 2022 | 4.18 | 0.10 | 3.05 | 6.72 | 0.59 | 10.43 |
| | Average | 4.28 | 0.05 | 1.47 | 5.85 | 0.36 | 8.52 |
| Harvester yield, t ha$^{-1}$ | 2021 | 3.94 | 010 | 2.82 | 4.56 | 0.23 | 4.46 |
| | 2022 | 3.67 | 0.06 | 2.19 | 5.62 | 0.26 | 5.39 |
| | Average | 3.80 | 0.05 | 1.41 | 5.09 | 0.20 | 4.63 |

In Trititrigia plants with a large number of grains in the ear 42.55 pcs, the low mass of one ear of 1.00 g and the mass of 1000 grains of 23.58 g were revealed, which indicates the fineness of the grain. According to these indicators, Trititrigia is inferior to winter wheat, which, with the number of grains in an ear of 29.02 pcs., has a high mass of one ear of

1.44 g and a mass of 1000 grains of 42.98 g, which demonstrates its advantage. The final indicator of cultivation is the combine yield. According to this indicator, Trititrigia is 25.34% inferior to winter wheat with its combine yield of 3.80 t ha$^{-1}$. Thus, a two-year trial of Trititrigia in the southern zone of the Rostov region showed low grain yield and high yield of straw mass.

### 3.2. Technological Indicators of Grain Quality

The results of determining the technological indicators of the Trititrigia grain quality in comparison with winter wheat are presented in Table 3.

**Table 3.** Technological indicators of the quality.

| Indicators | Years | Trititrigia "Pamyati Luybimovoy" | | | Winter Wheat "Stanichnaya" | | |
|---|---|---|---|---|---|---|---|
| | | Arithmetical Mean Value (Mean) | Average Linear Deviation (ALD) | Coefficient of Variation (CV) | Arithmetical Mean Value (Mean) | Average Linear Deviation (ALD) | Coefficient of Variation (CV) |
| Mass fraction of protein in grain, % | 2021 | 19.68 | 0.13 | 0.98 | 14.21 | 0.14 | 1.25 |
| | 2022 | 18.61 | 0.09 | 0.54 | 13.81 | 0.07 | 0.67 |
| | Average | 19.15 | 0.08 | 0.56 | 14.01 | 0.06 | 0.63 |
| The amount of gluten in the grain, % | 2021 | 34.51 | 0.23 | 0.82 | 29.61 | 0.13 | 0.57 |
| | 2022 | 32.18 | 0.15 | 0.58 | 30.18 | 0.07 | 0.26 |
| | Average | 33.34 | 0.19 | 0.70 | 29.90 | 0.04 | 0.19 |
| Gluten quality unit of the device | 2021 | 87.00 | 1.50 | 2.48 | 77.00 | 3.00 | 5.41 |
| | 2022 | 76.00 | 1.00 | 2.14 | 81.00 | 2.00 | 3.19 |
| | Average | 81.50 | 0.75 | 1.32 | 79.00 | 2.00 | 3.58 |
| Total vitreous of grain, % | 2021 | 67.00 | 2.50 | 4.72 | 50.00 | 2.00 | 4.90 |
| | 2022 | 61.00 | 1.00 | 2.32 | 54.00 | 1.50 | 3.38 |
| | Average | 64.00 | 1.50 | 3.06 | 52.00 | 0.25 | 0.79 |
| Number of drops, sec. | 2021 | 286 | 9.00 | 4.16 | 472 | 10.00 | 2.66 |
| | 2022 | 263 | 6.75 | 3.47 | 445 | 10.00 | 3.04 |
| | Average | 274 | 6.4 | 2.92 | 459 | 8.5 | 2.45 |

According to Table 2, it was found that, on average, in 2021 and 2022, the technical indicators of the quality of Trititrigia grain corresponded to high indicators. The mass fraction of protein 19.15% in Trititrigia grain was high and corresponded to the class of strong wheat (at least 16%), the amount of gluten 33.34% was high and corresponded to the class of strong wheat (at least 32.0%). The gluten deformation index (IDC) characterizing the quality of gluten was equal to 82 units of the device, which corresponded to the class of valuable wheat. The value of the total glassy grain (64%) corresponded to the class of strong wheat, and the number of falls is 274 s. Trititrigia corresponded to the class of strong wheat (at least 200 s). A low natural grain weight of 691 g L$^{-1}$ was noted (678 g L$^{-1}$ in 2021 and 704 g L$^{-1}$ in 2022). Due to the fact that the grain of Trititrigia was small, a low flour yield of 58.0–61.0% was noted.

### 3.3. Baking Properties

The baking properties of the flour obtained from the Trititrigia grain are presented in Table 4. Figure 5 shows the finished laboratory baking in the cutaway.

**Table 4.** Baking properties of Trititrigia and winter wheat.

| Indicators | Years | Trititrigia "Pamyati Luybimovoy" | | | Winter wheat "Stanichnaya" | | |
|---|---|---|---|---|---|---|---|
| | | Arithmetical Mean Value (Mean) | Average Linear Deviation (ALD) | Coefficient of Variation (CV) | Arithmetical Mean Value (Mean) | Arithmetical Mean Value (Mean) | Average Linear Deviation (ALD) |
| Specific work of deformation of the test, unit of the device | 2021 | 143.00 | 1.99 | 2.49 | 238.13 | 5.38 | 2.93 |
| | 2022 | 131.15 | 2.30 | 2.25 | 246.18 | 4.91 | 2.74 |
| | Average | 137.08 | 0.90 | 0.94 | 242.15 | 4.6 | 2.38 |
| Coefficient of the ratio of the elasticity of the dough to the extensibility | 2021 | 0.51 | 0.11 | 28.01 | 0.50 | 0.08 | 25.82 |
| | 2022 | 0.53 | 0.13 | 28.57 | 0.51 | 0.04 | 14.14 |
| | Average | 0.52 | 0.11 | 26.09 | 0.51 | 0.03 | 10.80 |
| Valorimetric estimation, valorimeter units | 2021 | 67.16 | 2.83 | 5.91 | 80.15 | 2.88 | 4.52 |
| | 2022 | 71.12 | 1.90 | 3.72 | 82.08 | 3.25 | 5.23 |
| | Average | 69.14 | 2.10 | 4.52 | 81.11 | 2.18 | 3.52 |
| Volumetric output of bread, cm$^3$ | 2021 | 630.20 | 10.10 | 2.23 | 760.02 | 19.33 | 3.05 |
| | 2022 | 600.23 | 16.58 | 3.25 | 810.07 | 8.62 | 1.34 |
| | Average | 615.21 | 12.89 | 2.56 | 785.04 | 8.91 | 1.48 |
| Bread shape, points | 2021 | 3.63 | 0.38 | 13.33 | 4.75 | 0.38 | 10.53 |
| | 2022 | 4.5 | 0.50 | 12.83 | 5.00 | 0.20 | 6.53 |
| | Average | 4.13 | 0.31 | 11.61 | 4.88 | 0.29 | 8.10 |
| Crumb porosity, points | 2021 | 1.53 | 0.23 | 18.06 | 4.05 | 0.25 | 9.13 |
| | 2022 | 1.40 | 0.10 | 11.66 | 4.08 | 0.33 | 9.89 |
| | Average | 1.46 | 0.12 | 11.64 | 4.06 | 0.26 | 8.61 |
| Crumb elasticity, points | 2021 | 2.06 | 0.36 | 22.34 | 4.14 | 0.16 | 5.27 |
| | 2022 | 2.15 | 0.18 | 11.38 | 4.27 | 0.19 | 6.31 |
| | Average | 2.11 | 0.19 | 12.40 | 4.20 | 0.11 | 3.73 |
| Total baking score, points | 2021 | 3.70 | 0.35 | 12.68 | 5.02 | 0.13 | 3.40 |
| | 2022 | 3.45 | 0.18 | 7.29 | 5.00 | 0.11 | 2.56 |
| | Average | 3.58 | 0.21 | 8.12 | 5.01 | 0.07 | 1.94 |

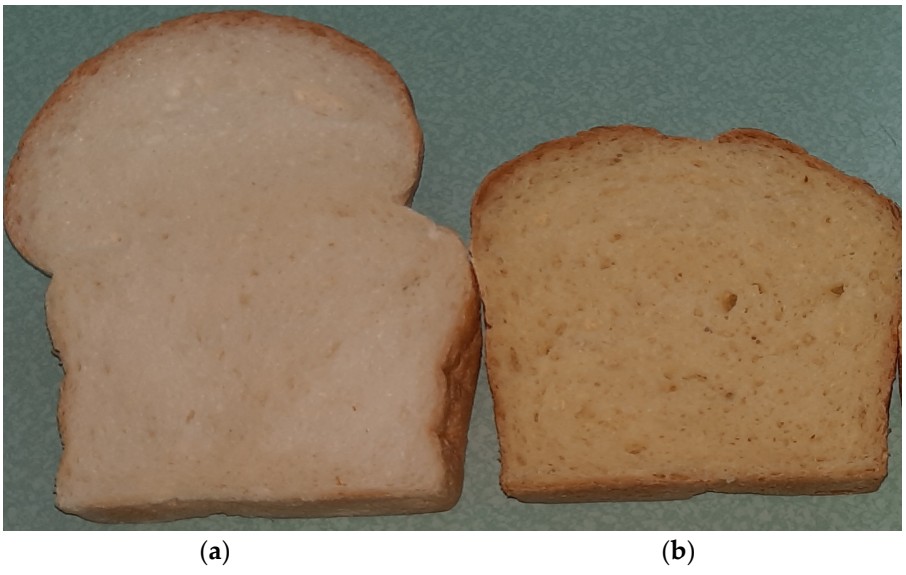

**Figure 5.** Winter wheat and Trititrigia baked in laboratory conditions: (**a**) winter wheat "Stanichnaya" and (**b**) Trititrigia "Pamyati Lyubimovoy".

As a result of the analysis of the rheological properties of the dough, it was found that the specific work of deformation of the dough (W) corresponded to the class of wheat fillers according to the classification standards used by the Central Laboratory of the State

Commission for Variety Testing of Agricultural Crops to characterize varieties in terms of grain quality and baking properties [47].

Coefficient of the ratio of dough elasticity to extensibility 0.52 also corresponded to the class of fillers.

The value of the valorimetric assessment (69.14 val. units) corresponded in quality to the class of valuable wheat (at least 55 val. units).

When assessing the quality of bread, a number of features are taken into account: shape of the bread (point), elasticity (point) and crumb porosity (point), volumetric yield of bread (mL). Main characteristics of bread quality are volumetric yield and overall baking grade, which in turn is an average indicator of shape of the bread, porosity, and elasticity of the crumb. Thus, the overall baking evaluation of the Trititrigia grain allowed it to be classified as a valuable wheat.

## 4. Discussion

Observations during the first test sowing of Trititrigia (2020–2021) showed that the abundant precipitation of the spring period of 2021 contributed to the intensive growth of plants. Observations during the second test sowing of Trititrigia (2021–2022) showed a delay in germination due to a lack of precipitation in September–October 2021, so the first shoots appeared in November. The earing phase of Trititrigia plants has been observed since mid-June. At the beginning of the first decade of July, the plants entered the flowering phase. The maturation of Trititrigia occurred in the last decade of July 2022.

The results of harvesting Trititrigia grain in the southern zone of the Rostov region showed a lower yield of grain with a high yield of straw mass compared to winter wheat "Stanichnaya". The low yield is presumably associated with the characteristics of the grain crop since the obtained yield data are similar to the data [36,37,44].

The results of the indicators of the elements of the crop structure, qualitative characteristics and baking properties of Trititrigia partially agree with the studies conducted by scientists [41,44] in the Moscow region: the yield of Trititrigia in the Rostov region was 3.8 t ha$^{-1}$, in the Moscow [36,37,44] 3.0–4.1 t ha$^{-1}$; the mass of 1000 grains in the Rostov region was 23.58 g, which is less than the grain of Trititrigia grown in the Moscow region [36,37,44]—31–35 g; the length of the ear and the height of the Trititrigia plant grown in the Moscow and Rostov regions coincide and amounted to 10–15 cm and 135–150 cm, respectively; the mass fraction of protein in the grain of Trititrigia grown in the Rostov region is higher and amounted to 19.15% than in Trititrigia grown in the Moscow region and was 17.5–18.4%; according to [36,37,44] the nature of Trititrigia grain is 774–800 g L$^{-1}$, according to the results of our research, the nature was 691 g L$^{-1}$; the baking score [41] was 3.58, within the framework of our study, the baking score was 3.70 with an average linear deviation of 0.21. Low porosity of the crumb (1.46 in Trititrigia and 4.06 in winter wheat "Stanichnaya" and low volume yield of bread (615.21 cm$^3$ in Trititrigia and 785.04 cm$^3$ in winter wheat "Stanichnaya") it is associated with a high protein content, which, by absorbing moisture, makes the dough more viscous. This is also reflected in the specific work of the dough deformation, which is lower in Trititrigia than in winter wheat "Stanichnaya" by 105 units. In addition, the low viscosity of the Trititrigia grain flour dough in comparison with the dough obtained from winter wheat flour "Stanichnaya" may be due to the composition of proteins that are different in wheatgrass. Thus, according to [43], the protein network of the test is mainly based on non-covalent interactions, as a result of which the proteins of the Trititrigia grain did not turn into a viscoelastic network.

After mowing, the observation of Trititrigia was continued, since, according to the breeders, the new crop was positioned as a perennial wheat with a sign of drought resistance [36,37,44]. However, according to the results of two-year tests, the vegetation of plants after mowing in 2021 and 2022 stopped completely. Partial death of the root system was observed even during the grain ripening phase. Insufficient precipitation and the effects of high temperatures in summer, which are characteristic of the southern zone of

the Rostov region, led to the death of the root system of Trititrigia. The growing season of Trititrigia turned out to be identical to winter wheat and corresponded to one year.

Requirements for the quality of grain for food purposes are prescribed in the national standard of the Russian Federation GOST 9353–2016 (Wheat. Specifications). In accordance with the requirements of this standard, the purchase and sale of grain lots is carried out. GOST 9353–2016 includes only six indicators (protein, amount of gluten, gluten deformation index (GDI), falling number, vitreousness, and nature). The quality class is awarded according to the lowest of the set of indicators.

Thus, in terms of the amount of protein and gluten, the number of falls, the results of the grain quality indicators of Trititrigia "Pamyati Lyubimovoy" corresponded to the first quality class, according to the GDI—to the third class, and according to the natural mass of the grain—to the fifth class. At the same time, all indicators of winter wheat "Stanichnaya" grain corresponded to the first class.

## 5. Conclusions

According to the results of a two-year test of Trititrigia in the semi-arid southern zone of the Rostov region, it was found that according to such economically valuable characteristics as yield, technological indicators of grain and baking properties of flour, this crop is adaptive for its cultivation as a classic winter crop, but the sign of drought resistance was not confirmed.

Despite the low yield (3.80 t ha$^{-1}$ vs. 5.62 t ha$^{-1}$), Trititrigia can find application in food production. In the conditions of the Rostov region, as stated in the register [39], Trititrigia has a high protein content of more than 19% and can be used in the preparation of functional foods. In the production of bakery products, due to the low porosity of the crumb and the volume of bread, it is advisable to conduct research on combining different types of flour in order to improve baking properties.

The high straw content of the bread mass (the ratio of straw/grain for Trititrigia and winter wheat "Stanichnaya" was 3.98 and 1.05, respectively), allows the use of Trititrigia not only for food, but also for fodder purposes.

Further research is also needed on the cultivation of Trititrigia in order to identify optimal conditions (tillage, fertilization, etc.), under which long-term wheat Trititrigia will yield in the second and third year after its sowing.

**Author Contributions:** Conceptualization, Y.L., B.M. and V.P.; methodology, Y.S., S.K. and T.M.; literature search, Y.S. and T.M.; formal analysis, Y.L., V.P. and S.K.; investigation, S.K., Y.S., D.R. and T.M.; resources, B.M., V.P. and S.K.; writing—review and editing, Y.S., D.R. and T.M.; supervision, Y.L., B.M., V.P., S.K. and D.R.; funding acquisition, V.P. and D.R. All authors have read and agreed to the published version of the manuscript.

**Funding:** This research was funded by Don State Technical University. The study was supported by a grant within the framework of the "Nauka–2030".

**Institutional Review Board Statement:** Not applicable.

**Informed Consent Statement:** Not applicable.

**Data Availability Statement:** Not applicable.

**Acknowledgments:** Authors of the current study thank this institution and its leader and author of the variety Upelniek V.P. for the provided breeding material and consulting assistance. The study was supported by a grant within the framework of the "Nauka–2030".

**Conflicts of Interest:** The authors declare no conflict of interest.

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
