# Peer review of "Experience in the Cultivation of a New Perennial Cereal Crop—Trititrigia in the Conditions of South of the Rostov Region"

_agriculture, doi:10.3390/agriculture13030605_

Round 1
Reviewer 1 Report
Dear Editor and Authors:
It is my pleasure to review this manuscript, which is an interesting study. The article presents the research results on cultivation of a new perennial cereal crop of winter wheat hybrid and wheatgrass – perennial winter wheat (Trititrigia) of the "Pamyati Lyubimovoy" variety (hereinafter – Trititrigia) in southern zone of the Rostov region. The study would improve the local wheat production and increased farmer income. However, current manuscript need major revise before publication.
There are many factors affecting the field test. Can the author show that the yield and wheat quality of this variety are better with only one year's field test? Field trials will take at least two years. If it meets the requirements of the publication, I have no problem with that.
The discussion is too short. The author should explain why this wheat variety has increased its yield. And explain why the quality of this wheat variety has improved. For example, is it because of increased photosynthesis and so on. Reference should be made to the relevant literature.
There are some small mistakes in the article, please proofread carefully. I will not list them all. A comma should be followed by a space!
Author Response
Dear Reviewer,
Thank you for the time given to this manuscript and valuable comments. We appreciate the informational help you provided; your comments allowed us to improve this manuscript.
Please see the attachment.

Reviewer 2 Report
Comments and suggestions for Authors
Experience in the cultivation of a new perennial cereal crop –Trititrigia in the conditions of south of the Rostov region
The presented manuscript deals with the current local problem. The aim of this study was to assess the suitability of using Trititrigia in the southern zone of the Rostov region. The subject is interesting and fall within the scope of the journal. The experimental dataset undoubtedly are useful and constitutes scientific values.
General remarks
The main comments are: one year of research and no statistical analysis of the results obtained.
The Abstract needs to be completed and corrected. The years of research, the name of the statistical method according to which the field experiment was conducted should be completed. It is also necessary to supplement the fertilization used in the cultivation of the test plant. Lines 20-21: Authors are the.... should be removed. Lines 27-29: Pre- sowing soil… should be removed.
Introduction: The introduction should be verified again, because some information is loosely related to the topic of the article. Lines 83-84: Acknowledgments should be placed at the end of the manuscript, before References. Please see other articles. ll References are in the Introduction. There are no references to the results obtained in the Discussion section. There is no research hypothesis.
M & M: What system was Trititrigia cultivated in? Zero or Traditional or..... (seeFigure 1). To be completed:
§ how many research factors have been taken into account,
§ years of research,
§ the size of the plots,
§ soil type according to WRB,
§ the name of the statistical method by which the experiment was established,
§ description of the principles of methods for the determination of humus, pH, available forms P and K,
§ what fertilization was used.
§ with the given content of available forms P and K, need to specify soil abundance.
The geographical coordinates of the research site or a map should be provided. Lines 130-141 should be removed. From what area were the plant samples taken? L192-193 should be removed.
What statistical methods were used to analyze the results?
Results: There is no statistical analysis in the data presentation (Tables 1-3).
Discussion: The Discussion section is not a discussion of the results. This is a continuation of the results description. Overall, there is no discussion of the results in the manuscript. All References are in the Introduction section.
Conclusions: The one-year results of the field experiment do not allow for correct conclusions. The presented conclusions repeat the description of the results.
Specific comments
Units should be adapted to editorial requirements in the entire manuscript.
The entire manuscript must be adapted to the publishing requirements.
Best regards
Author Response

(The authors gave the same response as above.)

Round 2
Reviewer 2 Report
Comments and suggestions for Authors
Experience in the cultivation of a new perennial cereal crop –Trititrigia in the conditions of south of the Rostov region
Dear Authors,
the manuscript has not been fully revised. The entire manuscript should be reviewed again and then carefully corrected and supplemented.
Comments
Lines 18-19 2020-2022 the years of research should be written correctly. Two or three years of research?
Figure 1 - it should be clearly stated that Trititrigia was cultivated in the traditional system.
Table 1 - make a table according to publishing requirements. Content total N – g kg-1 unit.
Lines 173-186 should be supplemented with the authors of the described analytical methods.
The title of subsections 2.2 and 3.1 and the title of Table 4 - please write in English.
Also in Table 4, English entries should be corrected.
Again, please pay attention to the correct spelling of references.
Author Response
Пожалуйста, посмотрите приложение.
